# Hyperbolic Neural Networks

**Octavian-Eugen Ganea**[*]
Dept. of Computer Science
ETH Zürich
Zurich, Switzerland

**Gary Bécigneul**[*]
Dept. of Computer Science
ETH Zürich
Zurich, Switzerland

**Thomas Hofmann**
Dept. of Computer Science
ETH Zürich
Zurich, Switzerland

## Abstract

Hyperbolic spaces have recently gained momentum in the context of machine learning due to their high capacity and tree-likeliness properties. However, the representational power of hyperbolic geometry is not yet on par with Euclidean geometry, mostly because of the absence of corresponding hyperbolic neural network layers. This makes it hard to use hyperbolic embeddings in downstream tasks. Here, we bridge this gap in a principled manner by combining the formalism of Möbius gyrovector spaces with the Riemannian geometry of the Poincaré model of hyperbolic spaces. As a result, we derive hyperbolic versions of important deep learning tools: multinomial logistic regression, feed-forward and recurrent neural networks such as gated recurrent units. This allows to embed sequential data and perform classification in the hyperbolic space. Empirically, we show that, even if hyperbolic optimization tools are limited, hyperbolic sentence embeddings either outperform or are on par with their Euclidean variants on textual entailment and noisy-prefix recognition tasks.

## 1 Introduction

It is common in machine learning to represent data as being embedded in the Euclidean space $\mathbb{R}^n$. The main reason for such a choice is simply convenience, as this space has a vectorial structure, closed-form formulas of distance and inner-product, and is the natural generalization of our intuition-friendly, visual three-dimensional space. Moreover, embedding entities in such a continuous space allows to feed them as input to neural networks, which has led to unprecedented performance on a broad range of problems, including sentiment detection [15], machine translation [3], textual entailment [22] or knowledge base link prediction [20, 6].

Despite the success of Euclidean embeddings, recent research has proven that many types of complex data (e.g. graph data) from a multitude of fields (e.g. Biology, Network Science, Computer Graphics or Computer Vision) exhibit a highly non-Euclidean latent anatomy [8]. In such cases, the Euclidean space does not provide the most powerful or meaningful geometrical representations. For example, [10] shows that arbitrary tree structures cannot be embedded with arbitrary low distortion (i.e. almost preserving their metric) in the Euclidean space with unbounded number of dimensions, but this task becomes surprisingly easy in the hyperbolic space with only 2 dimensions where the exponential growth of distances matches the exponential growth of nodes with the tree depth.

The adoption of neural networks and deep learning in these non-Euclidean settings has been rather limited until very recently, the main reason being the non-trivial or impossible principled generalizations of basic operations (e.g. vector addition, matrix-vector multiplication, vector translation, vector inner product) as well as, in more complex geometries, the lack of closed form expressions for basic objects (e.g. distances, geodesics, parallel transport). Thus, classic tools such as multinomial

---

[*]Equal contribution, correspondence at {`octavian.ganea,gary.becigneul`}@inf.ethz.ch

logistic regression (MLR), feed forward (FFNN) or recurrent neural networks (RNN) did not have a correspondence in these geometries.

*How should one generalize deep neural models to non-Euclidean domains ?* In this paper we address this question for one of the simplest, yet useful, non-Euclidean domains: spaces of constant negative curvature, i.e. *hyperbolic*. Their tree-likeness properties have been extensively studied [12, 13, 26] and used to visualize large taxonomies [18] or to embed heterogeneous complex networks [17]. In machine learning, recently, hyperbolic representations greatly outperformed Euclidean embeddings for hierarchical, taxonomic or entailment data [21, 10, 11]. Disjoint subtrees from the latent hierarchical structure surprisingly disentangle and cluster in the embedding space as a simple reflection of the space's negative curvature. However, appropriate deep learning tools are needed to embed feature data in this space and use it in downstream tasks. For example, implicitly hierarchical sequence data (e.g. textual entailment data, phylogenetic trees of DNA sequences or hierarchial captions of images) would benefit from suitable hyperbolic RNNs.

The **main contribution** of this paper is to bridge the gap between hyperbolic and Euclidean geometry in the context of neural networks and deep learning by generalizing in a principled manner both the basic operations as well as multinomial logistic regression (MLR), feed-forward (FFNN), simple and gated (GRU) recurrent neural networks (RNN) to the Poincaré model of the hyperbolic geometry. We do it by connecting the theory of gyrovector spaces and generalized Möbius transformations introduced by [2, 26] with the Riemannian geometry properties of the manifold. We smoothly parametrize basic operations and objects in all spaces of constant negative curvature using a unified framework that depends only on the curvature value. Thus, we show how Euclidean and hyperbolic spaces can be continuously deformed into each other. On a series of experiments and datasets we showcase the effectiveness of our hyperbolic neural network layers compared to their "classic" Euclidean variants on textual entailment and noisy-prefix recognition tasks. We hope that this paper will open exciting future directions in the nascent field of Geometric Deep Learning.

## 2    The Geometry of the Poincaré Ball

Basics of differential geometry are presented in appendix A.

### 2.1    Hyperbolic space: the Poincaré ball

The hyperbolic space has five isometric models that one can work with [9]. Similarly as in [21] and [11], we choose to work in the Poincaré ball. The *Poincaré ball model* $(\mathbb{D}^n, g^{\mathbb{D}})$ is defined by the manifold $\mathbb{D}^n = \{x \in \mathbb{R}^n : \|x\| < 1\}$ equipped with the following Riemannian metric:

$$g_x^{\mathbb{D}} = \lambda_x^2 g^E, \quad \text{where } \lambda_x := \frac{2}{1 - \|x\|^2}, \tag{1}$$

$g^E = \mathbf{I}_n$ being the Euclidean metric tensor. Note that the hyperbolic metric tensor is conformal to the Euclidean one. The *induced distance* between two points $x, y \in \mathbb{D}^n$ is known to be given by

$$d_{\mathbb{D}}(x, y) = \cosh^{-1}\left(1 + 2\frac{\|x - y\|^2}{(1 - \|x\|^2)(1 - \|y\|^2)}\right). \tag{2}$$

Since the Poincaré ball is conformal to Euclidean space, the *angle* between two vectors $u, v \in T_x\mathbb{D}^n \setminus \{\mathbf{0}\}$ is given by

$$\cos(\angle(u, v)) = \frac{g_x^{\mathbb{D}}(u, v)}{\sqrt{g_x^{\mathbb{D}}(u, u)}\sqrt{g_x^{\mathbb{D}}(v, v)}} = \frac{\langle u, v \rangle}{\|u\|\|v\|}. \tag{3}$$

### 2.2    Gyrovector spaces

In Euclidean space, natural operations inherited from the vectorial structure, such as vector addition, subtraction and scalar multiplication are often useful. The framework of *gyrovector spaces* provides an elegant non-associative algebraic formalism for hyperbolic geometry just as vector spaces provide the algebraic setting for Euclidean geometry [2, 25, 26].

In particular, these operations are used in special relativity, allowing to add speed vectors belonging to the Poincaré ball of radius $c$ (the celerity, *i.e.* the speed of light) so that they remain in the ball, hence not exceeding the speed of light.

We will make extensive use of these operations in our definitions of hyperbolic neural networks.

For $c \geq 0$, denote[2] by $\mathbb{D}_c^n := \{x \in \mathbb{R}^n \mid c\|x\|^2 < 1\}$. Note that if $c = 0$, then $\mathbb{D}_c^n = \mathbb{R}^n$; if $c > 0$, then $\mathbb{D}_c^n$ is the open ball of radius $1/\sqrt{c}$. If $c = 1$ then we recover the usual ball $\mathbb{D}^n$. Note that for $c, c' > 0$, $\mathbb{D}_c^n$ and $\mathbb{D}_{c'}^n$ are isometric.

**Möbius addition.**  The *Möbius addition* of $x$ and $y$ in $\mathbb{D}_c^n$ is defined as

$$x \oplus_c y := \frac{(1 + 2c\langle x, y\rangle + c\|y\|^2)x + (1 - c\|x\|^2)y}{1 + 2c\langle x, y\rangle + c^2\|x\|^2\|y\|^2}. \tag{4}$$

In particular, when $c = 0$, one recovers the Euclidean addition of two vectors in $\mathbb{R}^n$. Note that without loss of generality, the case $c > 0$ can be reduced to $c = 1$. Unless stated otherwise, we will use $\oplus$ as $\oplus_1$ to simplify notations. For general $c > 0$, this operation is not commutative nor associative. However, it satisfies $x \oplus_c \mathbf{0} = \mathbf{0} \oplus_c x = x$. Moreover, for any $x, y \in \mathbb{D}_c^n$, we have $(-x) \oplus_c x = x \oplus_c (-x) = \mathbf{0}$ and $(-x) \oplus_c (x \oplus_c y) = y$ (left-cancellation law). The *Möbius substraction* is then defined by the use of the following notation: $x \ominus_c y := x \oplus_c (-y)$. See [29, section 2.1] for a geometric interpretation of the Möbius addition.

**Möbius scalar multiplication.**  For $c > 0$, the *Möbius scalar multiplication* of $x \in \mathbb{D}_c^n \setminus \{\mathbf{0}\}$ by $r \in \mathbb{R}$ is defined as

$$r \otimes_c x := (1/\sqrt{c})\tanh(r\tanh^{-1}(\sqrt{c}\|x\|))\frac{x}{\|x\|}, \tag{5}$$

and $r \otimes_c \mathbf{0} := \mathbf{0}$. Note that similarly as for the Möbius addition, one recovers the Euclidean scalar multiplication when $c$ goes to zero: $\lim_{c \to 0} r \otimes_c x = rx$. This operation satisfies desirable properties such as $n \otimes_c x = x \oplus_c \cdots \oplus_c x$ ($n$ additions), $(r + r') \otimes_c x = r \otimes_c x \oplus_c r' \otimes_c x$ (scalar distributivity[3]), $(rr') \otimes_c x = r \otimes_c (r' \otimes_c x)$ (scalar associativity) and $|r| \otimes_c x/\|r \otimes_c x\| = x/\|x\|$ (scaling property).

**Distance.**  If one defines the generalized hyperbolic metric tensor $g^c$ as the metric conformal to the Euclidean one, with conformal factor $\lambda_x^c := 2/(1 - c\|x\|^2)$, then the induced distance function on $(\mathbb{D}_c^n, g^c)$ is given by[4]

$$d_c(x, y) = (2/\sqrt{c})\tanh^{-1}\left(\sqrt{c}\| - x \oplus_c y\|\right). \tag{6}$$

Again, observe that $\lim_{c \to 0} d_c(x, y) = 2\|x - y\|$, *i.e.* we recover Euclidean geometry in the limit[5]. Moreover, for $c = 1$ we recover $d_\mathbb{D}$ of Eq. (2).

**Hyperbolic trigonometry.**  Similarly as in the Euclidean space, one can define the notions of hyperbolic angles or *gyroangles* (when using the $\oplus_c$), as well as hyperbolic law of sines in the generalized Poincaré ball $(\mathbb{D}_c^n, g^c)$. We make use of these notions in our proofs. See Appendix B.

## 2.3  Connecting Gyrovector spaces and Riemannian geometry of the Poincaré ball

In this subsection, we present how geodesics in the Poincaré ball model are usually described with Möbius operations, and push one step further the existing connection between gyrovector spaces and the Poincaré ball by finding new identities involving the exponential map, and parallel transport.

In particular, these findings provide us with a simpler formulation of Möbius scalar multiplication, yielding a natural definition of matrix-vector multiplication in the Poincaré ball.

**Riemannian gyroline element.**  The Riemannian gyroline element is defined for an infinitesimal $dx$ as $ds := (x + dx) \ominus_c x$, and its size is given by [26, section 3.7]:

$$\|ds\| = \|(x + dx) \ominus_c x\| = \|dx\|/(1 - c\|x\|^2). \tag{7}$$

What is remarkable is that it turns out to be identical, up to a scaling factor of 2, to the usual line element $2\|dx\|/(1 - c\|x\|^2)$ of the Riemannian manifold $(\mathbb{D}_c^n, g^c)$.

**Geodesics.** The geodesic connecting points $x, y \in \mathbb{D}_c^n$ is shown in [2, 26] to be given by:

$$\gamma_{x \to y}(t) := x \oplus_c (-x \oplus_c y) \otimes_c t, \quad \text{with } \gamma_{x \to y} : \mathbb{R} \to \mathbb{D}_c^n \text{ s.t. } \gamma_{x \to y}(0) = x \text{ and } \gamma_{x \to y}(1) = y. \tag{8}$$

Note that when $c$ goes to $0$, geodesics become straight-lines, recovering Euclidean geometry. In the remainder of this subsection, we connect the gyrospace framework with Riemannian geometry.

**Lemma 1.** *For any $x \in \mathbb{D}^n$ and $v \in T_x \mathbb{D}_c^n$ s.t. $g_x^c(v, v) = 1$, the **unit-speed geodesic** starting from $x$ with direction $v$ is given by:*

$$\gamma_{x,v}(t) = x \oplus_c \left( \tanh \left( \sqrt{c} \frac{t}{2} \right) \frac{v}{\sqrt{c}\|v\|} \right), \quad \text{where } \gamma_{x,v} : \mathbb{R} \to \mathbb{D}^n \text{ s.t. } \gamma_{x,v}(0) = x \text{ and } \dot{\gamma}_{x,v}(0) = v. \tag{9}$$

*Proof.* One can use Eq. (8) and reparametrize it to unit-speed using Eq. (6). Alternatively, direct computation and identification with the formula in [11, Thm. 1] would give the same result. Using Eq. (6) and Eq. (9), one can sanity-check that $d_c(\gamma(0), \gamma(t)) = t, \forall t \in [0, 1]$. $\square$

**Exponential and logarithmic maps.** The following lemma gives the closed-form derivation of exponential and logarithmic maps.

**Lemma 2.** *For any point $x \in \mathbb{D}_c^n$, the exponential map $\exp_x^c : T_x \mathbb{D}_c^n \to \mathbb{D}_c^n$ and the logarithmic map $\log_x^c : \mathbb{D}_c^n \to T_x \mathbb{D}_c^n$ are given for $v \neq \mathbf{0}$ and $y \neq x$ by:*

$$\exp_x^c(v) = x \oplus_c \left( \tanh \left( \sqrt{c} \frac{\lambda_x^c \|v\|}{2} \right) \frac{v}{\sqrt{c}\|v\|} \right), \quad \log_x^c(y) = \frac{2}{\sqrt{c}\lambda_x^c} \tanh^{-1}(\sqrt{c}\| - x \oplus_c y\|) \frac{-x \oplus_c y}{\| - x \oplus_c y\|}. \tag{10}$$

*Proof.* Following the proof of [11, Cor. 1.1], one gets $\exp_x^c(v) = \gamma_{x, \frac{v}{\lambda_x^c \|v\|}}(\lambda_x^c \|v\|)$. Using Eq. (9) gives the formula for $\exp_x^c$. Algebraic check of the identity $\log_x^c(\exp_x^c(v)) = v$ concludes. $\square$

The above maps have more appealing forms when $x = \mathbf{0}$, namely for $v \in T_{\mathbf{0}} \mathbb{D}_c^n \setminus \{\mathbf{0}\}$, $y \in \mathbb{D}_c^n \setminus \{\mathbf{0}\}$:

$$\exp_{\mathbf{0}}^c(v) = \tanh(\sqrt{c}\|v\|) \frac{v}{\sqrt{c}\|v\|}, \quad \log_{\mathbf{0}}^c(y) = \tanh^{-1}(\sqrt{c}\|y\|) \frac{y}{\sqrt{c}\|y\|}. \tag{11}$$

Moreover, we still recover Euclidean geometry in the limit $c \to 0$, as $\lim_{c \to 0} \exp_x^c(v) = x + v$ is the Euclidean exponential map, and $\lim_{c \to 0} \log_x^c(y) = y - x$ is the Euclidean logarithmic map.

**Möbius scalar multiplication using exponential and logarithmic maps.** We studied the exponential and logarithmic maps in order to gain a better understanding of the Möbius scalar multiplication (Eq. (5)). We found the following:

**Lemma 3.** *The quantity $r \otimes x$ can actually be obtained by projecting $x$ in the tangent space at $\mathbf{0}$ with the logarithmic map, multiplying this projection by the scalar $r$ in $T_{\mathbf{0}} \mathbb{D}_c^n$, and then projecting it back on the manifold with the exponential map:*

$$r \otimes_c x = \exp_{\mathbf{0}}^c(r \log_{\mathbf{0}}^c(x)), \quad \forall r \in \mathbb{R}, x \in \mathbb{D}_c^n. \tag{12}$$

*In addition, we recover the well-known relation between geodesics connecting two points and the exponential map:*

$$\gamma_{x \to y}(t) = x \oplus_c (-x \oplus_c y) \otimes_c t = \exp_x^c(t \log_x^c(y)), \quad t \in [0, 1]. \tag{13}$$

This last result enables us to generalize scalar multiplication in order to define matrix-vector multiplication between Poincaré balls, one of the essential building blocks of hyperbolic neural networks.

**Parallel transport.** Finally, we connect parallel transport along the unique geodesic from $\mathbf{0}$ to $x$ to gyrovector spaces with the following theorem, which we prove in appendix C.

**Theorem 4.** *In the manifold $(\mathbb{D}_c^n, g^c)$, the parallel transport w.r.t. the Levi-Civita connection of a vector $v \in T_{\mathbf{0}} \mathbb{D}_c^n$ to another tangent space $T_x \mathbb{D}_c^n$ is given by the following isometry:*

$$P_{\mathbf{0} \to x}^c(v) = \log_x^c(x \oplus_c \exp_{\mathbf{0}}^c(v)) = \frac{\lambda_{\mathbf{0}}^c}{\lambda_x^c} v. \tag{14}$$

As we'll see later, this result is crucial in order to define and optimize parameters shared between different tangent spaces, such as biases in hyperbolic neural layers or parameters of hyperbolic MLR.

# 3 Hyperbolic Neural Networks

Neural networks can be seen as being made of compositions of basic operations, such as linear maps, bias translations, pointwise non-linearities and a final sigmoid or softmax layer. We first explain how to construct a softmax layer for logits lying in a Poincaré ball. Then, we explain how to transform a mapping between two Euclidean spaces as one between Poincaré balls, yielding matrix-vector multiplication and pointwise non-linearities in the Poincaré ball. Finally, we present possible adaptations of various recurrent neural networks to the hyperbolic domain.

## 3.1 Hyperbolic multiclass logistic regression

In order to perform multi-class classification on the Poincaré ball, one needs to generalize multinomial logistic regression (MLR) − also called softmax regression − to the Poincaré ball.

**Reformulating Euclidean MLR.** Let's first reformulate Euclidean MLR from the perspective of distances to margin hyperplanes, as in [19, Section 5]. This will allow us to easily generalize it.

Given $K$ classes, one learns a margin hyperplane for each such class using softmax probabilities:

$$\forall k \in \{1, ..., K\}, \quad p(y = k|x) \propto \exp\left((\langle a_k, x \rangle - b_k)\right), \quad \text{where } b_k \in \mathbb{R}, \ x, a_k \in \mathbb{R}^n. \quad (15)$$

Note that any affine hyperplane in $\mathbb{R}^n$ can be written with a normal vector $a$ and a scalar shift $b$:

$$H_{a,b} = \{x \in \mathbb{R}^n : \langle a, x \rangle - b = 0\}, \quad \text{where } a \in \mathbb{R}^n \setminus \{\mathbf{0}\}, \text{ and } b \in \mathbb{R}. \quad (16)$$

As in [19, Section 5], we note that $\langle a, x \rangle - b = \text{sign}(\langle a, x \rangle - b) \|a\| d(x, H_{a,b})$. Using Eq. (15):

$$p(y = k|x) \propto \exp(\text{sign}(\langle a_k, x \rangle - b_k) \|a_k\| d(x, H_{a_k, b_k})), \ b_k \in \mathbb{R}, x, a_k \in \mathbb{R}^n. \quad (17)$$

As it is not immediately obvious how to generalize the Euclidean hyperplane of Eq. (16) to other spaces such as the Poincaré ball, we reformulate it as follows:

$$\tilde{H}_{a,p} = \{x \in \mathbb{R}^n : \langle -p + x, a \rangle = 0\} = p + \{a\}^\perp, \text{ where } p \in \mathbb{R}^n, \ a \in \mathbb{R}^n \setminus \{\mathbf{0}\}. \quad (18)$$

This new definition relates to the previous one as $\tilde{H}_{a,p} = H_{a, \langle a, p \rangle}$. Rewriting Eq. (17) with $b = \langle a, p \rangle$:

$$p(y = k|x) \propto \exp(\text{sign}(\langle -p_k + x, a_k \rangle) \|a_k\| d(x, \tilde{H}_{a_k, p_k})), \text{ with } p_k, x, a_k \in \mathbb{R}^n. \quad (19)$$

It is now natural to adapt the previous definition to the hyperbolic setting by replacing $+$ by $\oplus_c$:

**Definition 3.1** (Poincaré hyperplanes). For $p \in \mathbb{D}_c^n$, $a \in T_p \mathbb{D}_c^n \setminus \{\mathbf{0}\}$, let $\{a\}^\perp := \{z \in T_p \mathbb{D}_c^n : g_p^c(z, a) = 0\} = \{z \in T_p \mathbb{D}_c^n : \langle z, a \rangle = 0\}$. Then, we define[6] Poincaré hyperplanes as

$$\tilde{H}_{a,p}^c := \{x \in \mathbb{D}_c^n : \langle \log_p^c(x), a \rangle_p = 0\} = \exp_p^c(\{a\}^\perp) = \{x \in \mathbb{D}_c^n : \langle -p \oplus_c x, a \rangle = 0\}. \quad (20)$$

The last equality is shown appendix D. $\tilde{H}_{a,p}^c$ can also be described as the union of images of all geodesics in $\mathbb{D}_c^n$ orthogonal to $a$ and containing $p$. Notice that our definition matches that of *hypergyroplanes*, see [27, definition 5.8]. A 3D hyperplane example is depicted in Fig. 1.

Next, we need the following theorem, proved in appendix E:

**Theorem 5.**

$$d_c(x, \tilde{H}_{a,p}^c) := \inf_{w \in \tilde{H}_{a,p}^c} d_c(x, w) = \frac{1}{\sqrt{c}} \sinh^{-1}\left(\frac{2\sqrt{c}|\langle -p \oplus_c x, a \rangle|}{(1 - c\| -p \oplus_c x\|^2)\|a\|}\right). \quad (21)$$

**Final formula for MLR in the Poincaré ball.** Putting together Eq. (19) and Thm. 5, we get the hyperbolic MLR formulation. Given $K$ classes and $k \in \{1, \ldots, K\}$, $p_k \in \mathbb{D}_c^n$, $a_k \in T_{p_k}\mathbb{D}_c^n \setminus \{\mathbf{0}\}$:

$$p(y = k|x) \propto \exp(\text{sign}(\langle -p_k \oplus_c x, a_k \rangle)\sqrt{g_{p_k}^c(a_k, a_k)}d_c(x, \tilde{H}_{a_k, p_k}^c)), \quad \forall x \in \mathbb{D}_c^n, \quad (22)$$

or, equivalently

$$p(y = k|x) \propto \exp\left(\frac{\lambda_{p_k}^c \|a_k\|}{\sqrt{c}} \sinh^{-1}\left(\frac{2\sqrt{c}\langle -p_k \oplus_c x, a_k\rangle}{(1 - c\| - p_k \oplus_c x\|^2)\|a_k\|}\right)\right), \quad \forall x \in \mathbb{D}_c^n. \tag{23}$$

Notice that when $c$ goes to zero, this goes to $p(y = k|x) \propto \exp(4\langle -p_k + x, a_k\rangle) = \exp((\lambda_{p_k}^0)^2\langle -p_k + x, a_k\rangle) = \exp(\langle -p_k + x, a_k\rangle_0)$, recovering the usual Euclidean softmax.

However, at this point it is unclear how to perform optimization over $a_k$, since it lives in $T_{p_k}\mathbb{D}_c^n$ and hence depends on $p_k$. The solution is that one should write $a_k = P_{\mathbf{0} \to p_k}^c(a_k') = (\lambda_{\mathbf{0}}^c/\lambda_{p_k}^c)a_k'$, where $a_k' \in T_{\mathbf{0}}\mathbb{D}_c^n = \mathbb{R}^n$, and optimize $a_k'$ as a Euclidean parameter.

## 3.2 Hyperbolic feed-forward layers

In order to define hyperbolic neural networks, it is crucial to define a canonically simple parametric family of transformations, playing the role of linear mappings in usual Euclidean neural networks, and to know how to apply pointwise non-linearities. Inspiring ourselves from our reformulation of Möbius scalar multiplication in Eq. (12), we define:

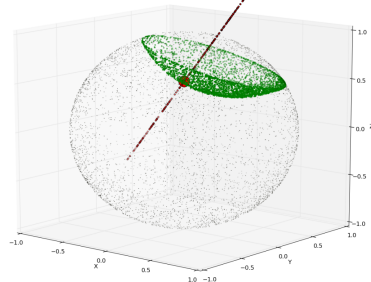

**Definition 3.2** (Möbius version). For $f : \mathbb{R}^n \to \mathbb{R}^m$, we define the *Möbius version of* $f$ as the map from $\mathbb{D}_c^n$ to $\mathbb{D}_c^m$ by:

$$f^{\otimes_c}(x) := \exp_{\mathbf{0}}^c(f(\log_{\mathbf{0}}^c(x))), \tag{24}$$

where $\exp_{\mathbf{0}}^c : T_{\mathbf{0}_m}\mathbb{D}_c^m \to \mathbb{D}_c^m$ and $\log_{\mathbf{0}}^c : \mathbb{D}_c^n \to T_{\mathbf{0}_n}\mathbb{D}_c^n$.

Figure 1: *An example of a hyperbolic hyperplane in $\mathbb{D}_1^3$ plotted using sampling. The red point is $p$. The shown normal axis to the hyperplane through $p$ is parallel to $a$.*

Note that similarly as for other Möbius operations, we recover the Euclidean mapping in the limit $c \to 0$ if $f$ is continuous, as $\lim_{c \to 0} f^{\otimes_c}(x) = f(x)$. This definition satisfies a few desirable properties too, such as: $(f \circ g)^{\otimes_c} = f^{\otimes_c} \circ g^{\otimes_c}$ for $f : \mathbb{R}^m \to \mathbb{R}^l$ and $g : \mathbb{R}^n \to \mathbb{R}^m$ (morphism property), and $f^{\otimes_c}(x)/\|f^{\otimes_c}(x)\| = f(x)/\|f(x)\|$ for $f(x) \neq \mathbf{0}$ (direction preserving). It is then straight-forward to prove the following result:

**Lemma 6** (Möbius matrix-vector multiplication). *If $M : \mathbb{R}^n \to \mathbb{R}^m$ is a linear map, which we identify with its matrix representation, then $\forall x \in \mathbb{D}_c^n$, if $Mx \neq \mathbf{0}$ we have*

$$M^{\otimes_c}(x) = (1/\sqrt{c})\tanh\left(\frac{\|Mx\|}{\|x\|}\tanh^{-1}(\sqrt{c}\|x\|)\right)\frac{Mx}{\|Mx\|}, \tag{25}$$

*and $M^{\otimes_c}(x) = \mathbf{0}$ if $Mx = \mathbf{0}$. Moreover, if we define the Möbius matrix-vector multiplication of $M \in \mathcal{M}_{m,n}(\mathbb{R})$ and $x \in \mathbb{D}_c^n$ by $M \otimes_c x := M^{\otimes_c}(x)$, then we have $(MM') \otimes_c x = M \otimes_c (M' \otimes_c x)$ for $M \in \mathcal{M}_{l,m}(\mathbb{R})$ and $M' \in \mathcal{M}_{m,n}(\mathbb{R})$ (matrix associativity), $(rM) \otimes_c x = r \otimes_c (M \otimes_c x)$ for $r \in \mathbb{R}$ and $M \in \mathcal{M}_{m,n}(\mathbb{R})$ (scalar-matrix associativity) and $M \otimes_c x = Mx$ for all $M \in \mathcal{O}_n(\mathbb{R})$ (rotations are preserved).*

**Pointwise non-linearity.** If $\varphi : \mathbb{R}^n \to \mathbb{R}^n$ is a pointwise non-linearity, then its Möbius version $\varphi^{\otimes_c}$ can be applied to elements of the Poincaré ball.

**Bias translation.** The generalization of a translation in the Poincaré ball is naturally given by moving along geodesics. But should we use the Möbius sum $x \oplus_c b$ with a hyperbolic bias $b$ or the exponential map $\exp_x^c(b')$ with a Euclidean bias $b'$? These views are unified with parallel transport (see Thm 4). Möbius translation of a point $x \in \mathbb{D}_c^n$ by a bias $b \in \mathbb{D}_c^n$ is given by

$$x \leftarrow x \oplus_c b = \exp_x^c(P_{\mathbf{0} \to x}^c(\log_{\mathbf{0}}^c(b))) = \exp_x^c\left(\frac{\lambda_{\mathbf{0}}^c}{\lambda_x^c}\log_{\mathbf{0}}^c(b)\right). \tag{26}$$

We recover Euclidean translations in the limit $c \to 0$. Note that bias translations play a particular role in this model. Indeed, consider multiple layers of the form $f_k(x) = \varphi_k(M_k x)$, each of which having Möbius version $f_k^{\otimes_c}(x) = \varphi_k^{\otimes_c}(M_k \otimes_c x)$. Then their composition can be re-written $f_k^{\otimes_c} \circ \cdots \circ f_1^{\otimes_c} = \exp_{\mathbf{0}}^c \circ f_k \circ \cdots \circ f_1 \circ \log_{\mathbf{0}}^c$. This means that these operations can essentially be performed in Euclidean space. Therefore, it is the interposition between those with the bias translation of Eq. (26) which differentiates this model from its Euclidean counterpart.

**Concatenation of multiple input vectors.** If a vector $x \in \mathbb{R}^{n+p}$ is the (vertical) concatenation of two vectors $x_1 \in \mathbb{R}^n$, $x_2 \in \mathbb{R}^p$, and $M \in \mathcal{M}_{m,n+p}(\mathbb{R})$ can be written as the (horizontal) concatenation of two matrices $M_1 \in \mathcal{M}_{m,n}(\mathbb{R})$ and $M_2 \in \mathcal{M}_{m,p}(\mathbb{R})$, then $Mx = M_1 x_1 + M_2 x_2$. We generalize this to hyperbolic spaces: if we are given $x_1 \in \mathbb{D}_c^n$, $x_2 \in \mathbb{D}_c^p$, $x = (x_1 \ x_2)^T \in \mathbb{D}_c^n \times \mathbb{D}_c^p$, and $M, M_1, M_2$ as before, then we define $M \otimes_c x := M_1 \otimes_c x_1 \oplus_c M_2 \otimes_c x_2$. Note that when $c$ goes to zero, we recover the Euclidean formulation, as $\lim_{c\to 0} M \otimes_c x = \lim_{c\to 0} M_1 \otimes_c x_1 \oplus_c M_2 \otimes_c x_2 = M_1 x_1 + M_2 x_2 = Mx$. Moreover, hyperbolic vectors $x \in \mathbb{D}_c^n$ can also be "concatenated" with real features $y \in \mathbb{R}$ by doing: $M \otimes_c x \oplus_c y \otimes_c b$ with learnable $b \in \mathbb{D}_c^m$ and $M \in \mathcal{M}_{m,n}(\mathbb{R})$.

### 3.3 Hyperbolic RNN

**Naive RNN.** A simple RNN can be defined by $h_{t+1} = \varphi(W h_t + U x_t + b)$ where $\varphi$ is a pointwise non-linearity, typically $\tanh$, sigmoid, ReLU, etc. This formula can be naturally generalized to the hyperbolic space as follows. For parameters $W \in \mathcal{M}_{m,n}(\mathbb{R})$, $U \in \mathcal{M}_{m,d}(\mathbb{R})$, $b \in \mathbb{D}_c^m$, we define:

$$h_{t+1} = \varphi^{\otimes_c}(W \otimes_c h_t \oplus_c U \otimes_c x_t \oplus_c b), \quad h_t \in \mathbb{D}_c^n, \ x_t \in \mathbb{D}_c^d. \tag{27}$$

Note that if inputs $x_t$'s are Euclidean, one can write $\tilde{x}_t := \exp_0^c(x_t)$ and use the above formula, since $\exp_{W\otimes_c h_t}^c(P_{0\to W\otimes_c h_t}^c(U x_t)) = W \otimes_c h_t \oplus_c \exp_0^c(U x_t) = W \otimes_c h_t \oplus_c U \otimes_c \tilde{x}_t$.

**GRU architecture.** One can also adapt the GRU architecture:

$$
\begin{aligned}
r_t &= \sigma(W^r h_{t-1} + U^r x_t + b^r), \qquad z_t = \sigma(W^z h_{t-1} + U^z x_t + b^z), \\
\tilde{h}_t &= \varphi(W(r_t \odot h_{t-1}) + U x_t + b), \quad h_t = (1 - z_t) \odot h_{t-1} + z_t \odot \tilde{h}_t,
\end{aligned}
\tag{28}
$$

where $\odot$ denotes pointwise product. First, how should we adapt the pointwise multiplication by a scaling gate? Note that the definition of the Möbius version (see Eq. (24)) can be naturally extended to maps $f : \mathbb{R}^n \times \mathbb{R}^p \to \mathbb{R}^m$ as $f^{\otimes_c} : (h, h') \in \mathbb{D}_c^n \times \mathbb{D}_c^p \mapsto \exp_0^c(f(\log_0^c(h), \log_0^c(h')))$. In particular, choosing $f(h, h') := \sigma(h) \odot h'$ yields[7] $f^{\otimes_c}(h, h') = \exp_0^c(\sigma(\log_0^c(h)) \odot \log_0^c(h')) = \mathrm{diag}(\sigma(\log_0^c(h))) \otimes_c h'$. Hence we adapt $r_t \odot h_{t-1}$ to $\mathrm{diag}(r_t) \otimes_c h_{t-1}$ and the reset gate $r_t$ to:

$$r_t = \sigma \log_0^c(W^r \otimes_c h_{t-1} \oplus_c U^r \otimes_c x_t \oplus_c b^r), \tag{29}$$

and similarly for the update gate $z_t$. Note that as the argument of $\sigma$ in the above is unbounded, $r_t$ and $z_t$ can a priori take values onto the full range $(0, 1)$. Now the intermediate hidden state becomes:

$$\tilde{h}_t = \varphi^{\otimes_c}((W\mathrm{diag}(r_t)) \otimes_c h_{t-1} \oplus_c U \otimes_c x_t \oplus b), \tag{30}$$

where Möbius matrix associativity simplifies $W \otimes_c (\mathrm{diag}(r_t) \otimes_c h_{t-1})$ into $(W\mathrm{diag}(r_t)) \otimes_c h_{t-1}$. Finally, we propose to adapt the update-gate equation as

$$h_t = h_{t-1} \oplus_c \mathrm{diag}(z_t) \otimes_c (-h_{t-1} \oplus_c \tilde{h}_t). \tag{31}$$

Note that when $c$ goes to zero, one recovers the usual GRU. Moreover, if $z_t = \mathbf{0}$ or $z_t = \mathbf{1}$, then $h_t$ becomes $h_{t-1}$ or $\tilde{h}_t$ respectively, similarly as in the usual GRU. This adaptation was obtained by adapting [24]: in this work, the authors re-derive the update-gate mechanism from a first principle called *time-warping invariance*. We adapted their derivation to the hyperbolic setting by using the notion of *gyroderivative* [4] and proving a *gyro-chain-rule* (see appendix F).

## 4 Experiments

**SNLI task and dataset.** We evaluate our method on two tasks. The first is natural language inference, or textual entailment. Given two sentences, a premise (e.g. "Little kids A. and B. are playing soccer.") and a hypothesis (e.g. "Two children are playing outdoors."), the binary classification task is to predict whether the second sentence can be inferred from the first one. This defines a partial order in the sentence space. We test hyperbolic networks on the biggest real dataset for this task, SNLI [7]. It consists of 570K training, 10K validation and 10K test sentence pairs. Following [28], we merge the "contradiction" and "neutral" classes into a single class of negative sentence pairs, while the "entailment" class gives the positive pairs.

| | SNLI | PREFIX-10% | PREFIX-30% | PREFIX-50% |
|---|---|---|---|---|
| FULLY EUCLIDEAN RNN | **79.34** % | 89.62 % | 81.71 % | 72.10 % |
| HYP RNN+FFNN, EUCL MLR | **79.18** % | 96.36 % | **87.83** % | **76.50** % |
| FULLY HYPERBOLIC RNN | 78.21 % | **96.91** % | 87.25 % | 62.94 % |
| FULLY EUCLIDEAN GRU | **81.52** % | 95.96 % | 86.47 % | 75.04 % |
| HYP GRU+FFNN, EUCL MLR | 79.76 % | **97.36** % | **88.47** % | **76.87** % |
| FULLY HYPERBOLIC GRU | **81.19** % | **97.14** % | **88.26** % | **76.44** % |

Table 1: Test accuracies for various models and four datasets. "Eucl" denotes Euclidean, "Hyp" denotes hyperbolic. All word and sentence embeddings have dimension 5. We highlight in **bold** the best baseline (or baselines, if the difference is less than 0.5%).

**PREFIX task and datasets.** We conjecture that the improvements of hyperbolic neural networks are more significant when the underlying data structure is closer to a tree. To test this, we design a proof-of-concept task of *detection of noisy prefixes*, i.e. given two sentences, one has to decide if the second sentence is a noisy prefix of the first, or a random sentence. We thus build synthetic datasets PREFIX-Z% (for Z being 10, 30 or 50) as follows: for each random first sentence of random length at most 20 and one random prefix of it, a second positive sentence is generated by randomly replacing Z% of the words of the prefix, and a second negative sentence of same length is randomly generated. Word vocabulary size is 100, and we generate 500K training, 10K validation and 10K test pairs.

Experimental details are presented in appendix G.

**Models architecture.** Our neural network layers can be used in a plug-n-play manner exactly like standard Euclidean layers. They can also be combined with Euclidean layers. However, optimization w.r.t. hyperbolic parameters is different (see below) and based on Riemannian gradients which are just rescaled Euclidean gradients when working in the conformal Poincaré model [21]. Thus, back-propagation can be applied in the standard way.

In our setting, we embed the two sentences using two distinct hyperbolic RNNs or GRUs. The sentence embeddings are then fed together with their squared distance (hyperbolic or Euclidean, depending on their geometry) to a FFNN (Euclidean or hyperbolic, see Sec. 3.2) which is further fed to an MLR (Euclidean or hyperbolic, see Sec. 3.1) that gives probabilities of the two classes (entailment vs neutral). We use cross-entropy loss on top. Note that hyperbolic and Euclidean layers can be mixed, e.g. the full network can be hyperbolic and only the last layer be Euclidean, in which case one has to use $\log_0$ and $\exp_0$ functions to move between the two manifolds in a correct manner as explained for Eq. 24. For the results shown in Tab. 1, we run each model (baseline or ours) exactly 3 times and report the test result corresponding to the best validation result from these 3 runs. We do this because the highly non-convex spectrum of hyperbolic neural networks sometimes results in convergence to poor local minima, suggesting that initialization is very important.

**Results.** Results are shown in Tab. 1. Note that the fully Euclidean baseline models might have an advantage over hyperbolic baselines because more sophisticated optimization algorithms such as Adam do not have a hyperbolic analogue at the moment. We first observe that all GRU models overpass their RNN variants. Hyperbolic RNNs and GRUs have the most significant improvement over their Euclidean variants when the underlying data structure is more tree-like, e.g. for PREFIX-10% − for which the tree relation between sentences and their prefixes is more prominent − we reduce the error by a factor of 3.35 for hyperbolic vs Euclidean RNN, and by a factor of 1.5 for hyperbolic vs Euclidean GRU. As soon as the underlying structure diverges more and more from a tree, the accuracy gap decreases − for example, for PREFIX-50% the noise heavily affects the representational power of hyperbolic networks. Also, note that on SNLI our methods perform similarly as with their Euclidean variants. Moreover, hyperbolic and Euclidean MLR are on par when used in conjunction with hyperbolic sentence embeddings, suggesting further empirical investigation is needed for this direction (see below).

**MLR classification experiments.** For the sentence entailment classification task we do not see a clear advantage of hyperbolic MLR compared to its Euclidean variant. A possible reason is that, when trained end-to-end, the model might decide to place positive and negative embeddings in a manner that is already well separated with a classic MLR. As a consequence, we

| WORDNET SUBTREE | MODEL | D = 2 | D = 3 | D = 5 | D = 10 |
|---|---|---|---|---|---|
| ANIMAL.N.01 3218 / 798 | HYP | **47.43 ± 1.07** | **91.92 ± 0.61** | **98.07 ± 0.55** | 99.26 ± 0.59 |
| | EUCL | 41.69 ± 0.19 | 68.43 ± 3.90 | 95.59 ± 1.18 | **99.36 ± 0.18** |
| | $\log_0$ | 38.89 ± 0.01 | 62.57 ± 0.61 | 89.21 ± 1.34 | 98.27 ± 0.70 |
| GROUP.N.01 6649 / 1727 | HYP | **81.72 ± 0.17** | **89.87 ± 2.73** | **87.89 ± 0.80** | **91.91 ± 3.07** |
| | EUCL | 61.13 ± 0.42 | 63.56 ± 1.22 | 67.82 ± 0.81 | **91.38 ± 1.19** |
| | $\log_0$ | 60.75 ± 0.24 | 61.98 ± 0.57 | 67.92 ± 0.74 | **91.41 ± 0.18** |
| WORKER.N.01 861 / 254 | HYP | **12.68 ± 0.82** | **24.09 ± 1.49** | **55.46 ± 5.49** | **66.83 ± 11.38** |
| | EUCL | 10.86 ± 0.01 | 22.39 ± 0.04 | 35.23 ± 3.16 | 47.29 ± 3.93 |
| | $\log_0$ | 9.04 ± 0.06 | 22.57 ± 0.20 | 26.47 ± 0.78 | 36.66 ± 2.74 |
| MAMMAL.N.01 953 / 228 | HYP | **32.01 ± 17.14** | **87.54 ± 4.55** | **88.73 ± 3.22** | **91.37 ± 6.09** |
| | EUCL | **15.58 ± 0.04** | 44.68 ± 1.87 | 59.35 ± 1.31 | 77.76 ± 5.08 |
| | $\log_0$ | 13.10 ± 0.13 | 44.89 ± 1.18 | 52.51 ± 0.85 | 56.11 ± 2.21 |

Table 2: Test F1 classification scores (%) for four different subtrees of WordNet noun tree. 95% confidence intervals for 3 different runs are shown for each method and each dimension. "Hyp" denotes our hyperbolic MLR, "Eucl" denotes directly applying Euclidean MLR to hyperbolic embeddings in their Euclidean parametrization, and $\log_0$ denotes applying Euclidean MLR in the tangent space at $\mathbf{0}$, after projecting all hyperbolic embeddings there with $\log_0$.

further investigate MLR for the task of subtree classification. Using an open source implementation[8] of [21], we pre-trained Poincaré embeddings of the WordNet noun hierarchy (82,115 nodes). We then choose one node in this tree (see Table 2) and classify all other nodes (solely based on their embeddings) as being part of the subtree rooted at this node. All nodes in such a subtree are divided into positive training nodes (80%) and positive test nodes (20%).

The same splitting procedure is applied for the remaining WordNet nodes that are divided into a negative training and negative test set respectively. Three variants of MLR are then trained on top of pre-trained Poincaré embeddings[21] to solve this binary classification task: hyperbolic MLR, Euclidean MLR applied directly on the hyperbolic embeddings (even if mathematically this is not respecting the hyperbolic geometry) and Euclidean MLR applied after mapping all embeddings in the tangent space at $\mathbf{0}$ using the $\log_0$ map. More experimental details in appendix G.2. Quantitative results are presented in Table 2. We can see that the hyperbolic MLR overpasses its Euclidean variants in almost all settings, sometimes by a large margin. Moreover, to provide further understanding, we plot the 2-dimensional embeddings and the trained separation hyperplanes (geodesics in this case) in Figure 2.

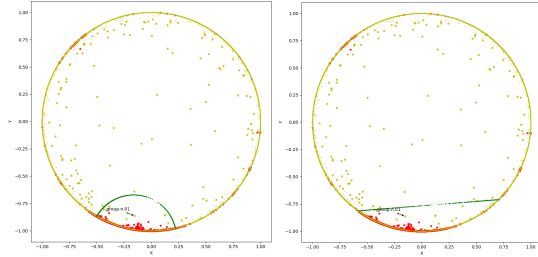

Figure 2: *Hyperbolic (left) vs Direct Euclidean (right) binary MLR used to classify nodes as being part in the* GROUP.N.01 *subtree of the WordNet noun hierarchy solely based on their Poincaré embeddings. The positive points (from the subtree) are in red, the negative points (the rest) are in yellow and the trained positive separation hyperplane is depicted in green.*

## 5 Conclusion

We showed how classic Euclidean deep learning tools such as MLR, FFNNs, RNNs or GRUs can be generalized in a principled manner to all spaces of constant negative curvature combining Riemannian geometry with the elegant theory of gyrovector spaces. Empirically we found that our models outperform or are on par with corresponding Euclidean architectures on sequential data with implicit hierarchical structure. We hope to trigger exciting future research related to better understanding of the hyperbolic non-convexity spectrum and development of other non-Euclidean deep learning methods. Our data and Tensorflow [1] code are publicly available[9].

## Acknowledgements

We thank Igor Petrovski for useful pointers regarding the implementation.

This research is funded by the Swiss National Science Foundation (SNSF) under grant agreement number 167176. Gary Bécigneul is also funded by the Max Planck ETH Center for Learning Systems.

## Footnotes

[2]We take different notations as in [25] where the author uses $s = 1/\sqrt{c}$.

[3]$\otimes_c$ has priority over $\oplus_c$ in the sense that $a \otimes_c b \oplus_c c := (a \otimes_c b) \oplus_c c$ and $a \oplus_c b \otimes_c c := a \oplus_c (b \otimes_c c)$.

[4]The notation $-x \oplus_c y$ should always be read as $(-x) \oplus_c y$ and not $-(x \oplus_c y)$.

[5]The factor 2 comes from the conformal factor $\lambda_x = 2/(1 - \|x\|^2)$, which is a convention setting the curvature to $-1$.

[6]where $\langle \cdot, \cdot \rangle$ denotes the (Euclidean) inner-product of the ambient space.

[7]If $x$ has $n$ coordinates, then $\mathrm{diag}(x)$ denotes the diagonal matrix of size $n$ with $x_i$'s on its diagonal.

[8] https://github.com/dalab/hyperbolic_cones

[9] https://github.com/dalab/hyperbolic_nn

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
