[Supplementary Material]

## A  Basics of Riemannian geometry

We briefly introduce basic concepts of differential geometry largely needed for a principled generalization of Euclidean neural networks. For more rigorous and in-depth expositions, see [23, 14].

An $n$-*dimensional manifold* $\mathcal{M}$ is a space that can locally be approximated by $\mathbb{R}^n$: it is a generalization to higher dimensions of the notion of a 2D surface. For $x \in \mathcal{M}$, one can define the *tangent space* $T_x\mathcal{M}$ of $\mathcal{M}$ at $x$ as the first order linear approximation of $\mathcal{M}$ around $x$. A *Riemannian metric* $g = (g_x)_{x \in \mathcal{M}}$ on $\mathcal{M}$ is a collection of inner-products $g_x : T_x\mathcal{M} \times T_x\mathcal{M} \to \mathbb{R}$ varying smoothly with $x$. A *Riemannian manifold* $(\mathcal{M}, g)$ is a manifold $\mathcal{M}$ equipped with a Riemannian metric $g$. Although a choice of a Riemannian metric $g$ on $\mathcal{M}$ only seems to define the geometry locally on $\mathcal{M}$, it induces global distances by integrating the length (of the speed vector living in the tangent space) of a shortest path between two points:

$$d(x, y) = \inf_{\gamma} \int_0^1 \sqrt{g_{\gamma(t)}(\dot{\gamma}(t), \dot{\gamma}(t))} dt, \tag{32}$$

where $\gamma \in \mathcal{C}^\infty([0, 1], \mathcal{M})$ is such that $\gamma(0) = x$ and $\gamma(1) = y$. A *geodesic* $\gamma$ is a smooth path of locally minimal length, and can be seen as the generalization of a straight-line in Euclidean space. In certain spaces, such as the hyperbolic space, there is a unique geodesic between two points, which allows to consider the *parallel transport from $x$ to $y$* (implicitly taken along this unique geodesic) $P_{x \to y} : T_x M \to T_y M$, which is a linear isometry between tangent spaces corresponding to moving tangent vectors along geodesics and defines a canonical way to connect tangent spaces. The *exponential map* $\exp_x$ at $x$, when well-defined, gives a way to project back a vector $v$ of the tangent space $T_x\mathcal{M}$ at $x$, to a point $\exp_x(v) \in \mathcal{M}$ on the manifold. This map is often used to parametrize a geodesic $\gamma$ starting from $\gamma(0) := x \in \mathcal{M}$ with unit-norm direction $\dot{\gamma}(0) := v \in T_x\mathcal{M}$ as $t \mapsto \exp_x(tv)$. For *geodesically complete manifolds*, such as the Poincaré ball considered in this work, $\exp_x$ is well-defined on the full tangent space $T_x\mathcal{M}$. Finally, a metric $\tilde{g}$ is said to be *conformal* to another metric $g$ if it defines the same angles, *i.e.*

$$\frac{\tilde{g}_x(u, v)}{\sqrt{\tilde{g}_x(u, u)}\sqrt{\tilde{g}_x(v, v)}} = \frac{g_x(u, v)}{\sqrt{g_x(u, u)}\sqrt{g_x(v, v)}}, \tag{33}$$

for all $x \in \mathcal{M}$, $u, v \in T_x\mathcal{M} \setminus \{\mathbf{0}\}$. This is equivalent to the existence of a smooth function $\lambda : \mathcal{M} \to \mathbb{R}$, called the *conformal factor*, such that $\tilde{g}_x = \lambda_x^2 g_x$ for all $x \in \mathcal{M}$.

## B  Hyperbolic Trigonometry

**Hyperbolic angles.**  For $A, B, C \in \mathbb{D}_c^n$, we denote by $\angle A := \angle BAC$ the angle between the two geodesics starting from $A$ and ending at $B$ and $C$ respectively. This angle can be defined in two equivalent ways: i) either using the angle between the initial velocities of the two geodesics as given by Eq. 3, or ii) using the formula

$$\cos(\angle A) = \left\langle \frac{(-A) \oplus_c B}{\|(-A) \oplus_c B\|}, \frac{(-A) \oplus_c C}{\|(-A) \oplus_c C\|} \right\rangle, \tag{34}$$

In this case, $\angle A$ is also called a *gyroangle* in the work of [26, section 4].

**Hyperbolic law of sines.**  We state here the hyperbolic law of sines. If for $A, B, C \in \mathbb{D}_c^n$, we denote by $\angle B := \angle ABC$ the angle between the two geodesics starting from $B$ and ending at $A$ and $C$ respectively, and by $\tilde{c} = d_c(B, A)$ the length of the hyperbolic segment BA (and similarly for others), then we have:

$$\frac{\sin(\angle A)}{\sinh(\sqrt{c}\tilde{a})} = \frac{\sin(\angle B)}{\sinh(\sqrt{c}\tilde{b})} = \frac{\sin(\angle C)}{\sinh(\sqrt{c}\tilde{c})}. \tag{35}$$

Note that one can also adapt the hyperbolic law of cosines to the hyperbolic space.

## C  Proof of Theorem 4

**Theorem 4.**
*In the manifold $(\mathbb{D}_c^n, g^c)$, the parallel transport w.r.t. the Levi-Civita connection of a vector $v \in T_\mathbf{0}\mathbb{D}_c^n$ to another tangent space $T_x\mathbb{D}_c^n$ is given by the following isometry:*

$$P_{\mathbf{0}\to x}^c(v) = \log_x^c(x \oplus_c \exp_\mathbf{0}^c(v)) = \frac{\lambda_\mathbf{0}^c}{\lambda_x^c}v. \tag{36}$$

*Proof.* The geodesic in $\mathbb{D}_c^n$ from $\mathbf{0}$ to $x$ is given in Eq. (8) by $\gamma(t) = x \otimes_c t$, for $t \in [0,1]$. Let $v \in T_\mathbf{0}\mathbb{D}_c^n$. Then it is of common knowledge that there exists a unique parallel[10] vector field $X$ along $\gamma$ (i.e. $X(t) \in T_{\gamma(t)}\mathbb{D}_c^n$, $\forall t \in [0,1]$) such that $X(0) = v$. Let's define:

$$X : t \in [0,1] \mapsto \log_{\gamma(t)}^c(\gamma(t) \oplus_c \exp_\mathbf{0}^c(v)) \in T_{\gamma(t)}\mathbb{D}_c^n. \tag{37}$$

Clearly, $X$ is a vector field along $\gamma$ such that $X(0) = v$. Now define

$$P_{\mathbf{0}\to x}^c : v \in T_\mathbf{0}\mathbb{D}_c^n \mapsto \log_x^c(x \oplus_c \exp_\mathbf{0}^c(v)) \in T_x\mathbb{D}_c^n. \tag{38}$$

From Eq. (10), it is easily seen that $P_{\mathbf{0}\to x}^c(v) = \frac{\lambda_\mathbf{0}^c}{\lambda_x^c}v$, hence $P_{\mathbf{0}\to x}^c$ is a linear isometry from $T_\mathbf{0}\mathbb{D}_c^n$ to $T_x\mathbb{D}_c^n$. Since $P_{\mathbf{0}\to x}^c(v) = X(1)$, it is enough to prove that $X$ is parallel in order to guarantee that $P_{\mathbf{0}\to x}^c$ is the parallel transport from $T_\mathbf{0}\mathbb{D}_c^n$ to $T_x\mathbb{D}_c^n$.

Since $X$ is a vector field along $\gamma$, its covariant derivative can be expressed with the Levi-Civita connection $\nabla^c$ associated to $g^c$:

$$\frac{DX}{\partial t} = \nabla_{\dot{\gamma}(t)}^c X. \tag{39}$$

Let's compute the Levi-Civita connection from its Christoffel symbols. In a local coordinate system, they can be written as

$$\Gamma_{jk}^i = \frac{1}{2}(g^c)^{il}(\partial_j g_{lk}^c + \partial_k g_{lj}^c - \partial_l g_{jk}^c), \tag{40}$$

where superscripts denote the inverse metric tensor and using Einstein's notations. As $g_{ij}^c = (\lambda^c)^2\delta_{ij}$, at $\gamma(t) \in \mathbb{D}_c^n$ this yields:

$$\Gamma_{jk}^i = c\lambda_{\gamma(t)}^c(\delta_{ik}\gamma(t)_j + \delta_{ij}\gamma(t)_k - \delta_{jk}\gamma(t)_i). \tag{41}$$

On the other hand, since $X(t) = (\lambda_\mathbf{0}^c/\lambda_{\gamma(t)}^c)v$, we have

$$\nabla_{\dot{\gamma}(t)}^c X = \dot{\gamma}(t)^i \nabla_i^c X = \dot{\gamma}(t)^i \nabla_i^c\left(\frac{\lambda_\mathbf{0}^c}{\lambda_{\gamma(t)}^c}v\right) = v^j \dot{\gamma}(t)^i \nabla_i^c\left(\frac{\lambda_\mathbf{0}^c}{\lambda_{\gamma(t)}^c}e_j\right). \tag{42}$$

Since $\gamma(t) = (1/\sqrt{c})\tanh(t\tanh^{-1}(\sqrt{c}\|x\|))\frac{x}{\|x\|}$, it is easily seen that $\dot{\gamma}(t)$ is colinear to $\gamma(t)$. Hence there exists $K_t^x \in \mathbb{R}$ such that $\dot{\gamma}(t) = K_t^x\gamma(t)$. Moreover, we have the following Leibniz rule:

$$\nabla_i^c\left(\frac{\lambda_\mathbf{0}^c}{\lambda_{\gamma(t)}^c}e_j\right) = \frac{\lambda_\mathbf{0}^c}{\lambda_{\gamma(t)}^c}\nabla_i^c e_j + \frac{\partial}{\partial\gamma(t)_i}\left(\frac{\lambda_\mathbf{0}^c}{\lambda_{\gamma(t)}^c}\right)e_j. \tag{43}$$

Combining these yields

$$\frac{DX}{\partial t} = K_t^x v^j \gamma(t)^i\left(\frac{\lambda_\mathbf{0}^c}{\lambda_{\gamma(t)}^c}\nabla_i^c e_j + \frac{\partial}{\partial\gamma(t)_i}\left(\frac{\lambda_\mathbf{0}^c}{\lambda_{\gamma(t)}^c}\right)e_j\right). \tag{44}$$

Replacing with the Christoffel symbols of $\nabla^c$ at $\gamma(t)$ gives

$$\frac{\lambda_\mathbf{0}^c}{\lambda_{\gamma(t)}^c}\nabla_i^c e_j = \frac{\lambda_\mathbf{0}^c}{\lambda_{\gamma(t)}^c}\Gamma_{ij}^k e_k = 2c[\delta_j^k\gamma(t)_i + \delta_i^k\gamma(t)_j - \delta_{ij}\gamma(t)^k]e_k. \tag{45}$$

Moreover,

$$\frac{\partial}{\partial\gamma(t)_i}\left(\frac{\lambda_\mathbf{0}^c}{\lambda_{\gamma(t)}^c}\right)e_j = \frac{\partial}{\partial\gamma(t)_i}\left(-c\|\gamma(t)\|^2\right)e_j = -2c\gamma(t)_i e_j. \tag{46}$$

Putting together everything, we obtain

$$\frac{DX}{\partial t} = K_t^x v^j \gamma(t)^i \left(2c[\delta_j^k \gamma(t)_i + \delta_i^k \gamma(t)_j - \delta_{ij}\gamma(t)^k]e_k - 2c\gamma(t)_i e_j\right) \tag{47}$$

$$= 2cK_t^x v^j \gamma(t)^i \left(\gamma(t)_j e_i - \delta_{ij}\gamma(t)^k e_k\right) \tag{48}$$

$$= 2cK_t^x v^j \left(\gamma(t)_j \gamma(t)^i e_i - \gamma(t)^i \delta_{ij}\gamma(t)^k e_k\right) \tag{49}$$

$$= 2cK_t^x v^j \left(\gamma(t)_j \gamma(t)^i e_i - \gamma(t)_j \gamma(t)^k e_k\right) \tag{50}$$

$$= 0, \tag{51}$$

which concludes the proof. $\qquad\square$

## D  Proof of Eq. (20)

*Proof.* Two steps proof:

*i)* $\exp_p^c(\{a\}^\perp) \subseteq \{x \in \mathbb{D}_c^n : \langle -p \oplus_c x, a \rangle = 0\}$:

Let $z \in \{a\}^\perp$. From Eq. (10), we have that:

$$\exp_p^c(z) = -p \oplus_c \beta z, \quad \text{for some } \beta \in \mathbb{R}. \tag{52}$$

This, together with the left-cancellation law in gyrospaces (see section 2.2), implies that

$$\langle -p \oplus_c \exp_p^c(z), a \rangle = \langle \beta z, a \rangle = 0 \tag{53}$$

which is what we wanted.

*ii)* $\{x \in \mathbb{D}_c^n : \langle -p \oplus_c x, a \rangle = 0\} \subseteq \exp_p^c(\{a\}^\perp)$:

Let $x \in \mathbb{D}_c^n$ s.t. $\langle -p \oplus_c x, a \rangle = 0$. Then, using Eq. (10), we derive that:

$$\log_p^c(x) = \beta(-p \oplus_c x), \quad \text{for some } \beta \in \mathbb{R}, \tag{54}$$

which is orthogonal to $a$, by assumption. This implies $\log_p^c(x) \in \{a\}^\perp$, hence $x \in \exp_p^c(\{a\}^\perp)$. $\quad\square$

## E  Proof of Theorem 5

**Theorem 5.**

$$d_c(x, \tilde{H}_{a,p}^c) := \inf_{w \in \tilde{H}_{a,p}^c} d_c(x, w) = \frac{1}{\sqrt{c}} \sinh^{-1} \left( \frac{2\sqrt{c}|\langle -p \oplus_c x, a \rangle|}{(1 - c\| - p \oplus_c x\|^2)\|a\|} \right). \tag{55}$$

*Proof.* We first need to prove the following lemma, trivial in the Euclidean space, but not in the Poincaré ball:

**Lemma 7.** *(Orthogonal projection on a geodesic) Any point in the Poincaré ball has a unique orthogonal projection on any given geodesic that does not pass through the point. Formally, for all $y \in \mathbb{D}_c^n$ and for all geodesics $\gamma_{x \to z}(\cdot)$ s.t. $y \notin Im\, \gamma_{x \to z}$, there exists an unique $w \in Im\, \gamma_{x \to z}$ s.t. $\angle(\gamma_{w \to y}, \gamma_{x \to z}) = \pi/2$.*

*Proof.* We first note that any geodesic in $\mathbb{D}_c^n$ has the form $\gamma(t) = u \oplus_c v \otimes_c t$ as given by Eq. 9, and has two "points at infinity" lying on the ball border ($v \neq \mathbf{0}$):

$$\gamma(\pm\infty) = u \oplus_c \frac{\pm v}{\sqrt{c}\|v\|} \in \partial\mathbb{D}_c^n. \tag{56}$$

Using the notations in the lemma statement, the closed-form of $\gamma_{x \to z}$ is given by Eq. (8):

$$\gamma_{x \to z}(t) = x \oplus_c (-x \oplus_c z) \otimes_c t$$

We denote by $x', z' \in \partial \mathbb{D}_c^n$ its points at infinity as described by Eq. (56). Then, the hyperbolic angle $\angle ywx'$ is well defined from Eq. (34):

$$\cos(\angle(\gamma_{w\to y}, \gamma_{x\to z})) = \cos(\angle ywz') = \frac{\langle -w \oplus_c y, -w \oplus_c z' \rangle}{\| -w \oplus_c y \| \cdot \| -w \oplus_c z' \|}. \tag{57}$$

We now perform 2 steps for this proof.

*i) Existence of $w$:*

The angle function from Eq. (57) is continuous w.r.t $t$ when $w = \gamma_{x\to z}(t)$. So we first prove existence of an angle of $\pi/2$ by continuously moving $w$ from $x'$ to $z'$ when $t$ goes from $-\infty$ to $\infty$, and observing that $\cos(\angle ywz')$ goes from $-1$ to $1$ as follows:

$$\cos(\angle yx'z') = 1 \quad \& \quad \lim_{w\to z'} \cos(\angle ywz') = -1. \tag{58}$$

The left part of Eq. (58) follows from Eq. (57) and from the fact (easy to show from the definition of $\oplus_c$) that $a \oplus_c b = a$, when $\|a\| = 1/\sqrt{c}$ (which is the case of $x'$). The right part of Eq. (58) follows from the fact that $\angle ywz' = \pi - \angle ywx'$ (from the conformal property, or from Eq. (34)) and $\cos(\angle yz'x') = 1$ (proved as above).

Hence $\cos(\angle ywz')$ has to pass through $0$ when going from $-1$ to $1$, which achieves the proof of existence.

*ii) Uniqueness of $w$:*

Assume by contradiction that there are two $w$ and $w'$ on $\gamma_{x\to z}$ that form angles $\angle ywx'$ and $\angle yw'x'$ of $\pi/2$. Since $w, w', x'$ are on the same geodesic, we have

$$\pi/2 = \angle yw'x' = \angle yw'w = \angle ywx' = \angle yw'w \tag{59}$$

So $\triangle yww'$ has two right angles, but in the Poincaré ball this is impossible. □

Now, we need two more lemmas:

**Lemma 8.** *(Minimizing distance from point to geodesic) The orthogonal projection of a point to a geodesic (not passing through the point) is minimizing the distance between the point and the geodesic.*

*Proof.* The proof is similar with the Euclidean case and it's based on hyperbolic sine law and the fact that in any right hyperbolic triangle the hypotenuse is strictly longer than any of the other sides. □

**Lemma 9.** *(Geodesics through $p$) Let $\tilde{H}_{a,p}^c$ be a Poincaré hyperplane. Then, for any $w \in \tilde{H}_{a,p}^c \setminus \{p\}$, all points on the geodesic $\gamma_{p\to w}$ are included in $\tilde{H}_{a,p}^c$.*

*Proof.* $\gamma_{p\to w}(t) = p \oplus_c (-p \oplus_c w) \otimes_c t$. Then, it is easy to check the condition in Eq. (20):

$$\langle -p \oplus_c \gamma_{p\to w}(t), a \rangle = \langle (-p \oplus_c w) \otimes_c t, a \rangle \propto \langle -p \oplus_c w, a \rangle = 0. \tag{60}$$

□

We now turn back to our proof. Let $x \in \mathbb{D}_c^n$ be an arbitrary point and $\tilde{H}_{a,p}^c$ a Poincaré hyperplane. We prove that there is at least one point $w^* \in \tilde{H}_{a,p}^c$ that achieves the infimum distance

$$d_c(x, w^*) = \inf_{w \in \tilde{H}_{a,p}^c} d_c(x, w), \tag{61}$$

and, moreover, that this distance is the same as the one in the theorem's statement.

We first note that for any point $w \in \tilde{H}_{a,p}^c$, if $\angle xwp \neq \pi/2$, then $w \neq w^*$. Indeed, using Lemma 8 and Lemma 9, it is obvious that the projection of $x$ to $\gamma_{p\to w}$ will give a strictly lower distance.

Thus, we only consider $w \in \tilde{H}_{a,p}^c$ such that $\angle xwp = \pi/2$. Applying hyperbolic sine law in the right triangle $\triangle xwp$, one gets:

$$d_c(x, w) = (1/\sqrt{c}) \sinh^{-1} \left( \sinh(\sqrt{c}\, d_c(x, p)) \cdot \sin(\angle xpw) \right). \tag{62}$$

One of the above quantities does not depend on $w$:

$$\sinh(\sqrt{c}\, d_c(x,p)) = \sinh(2\tanh^{-1}(\sqrt{c}\| - p \oplus_c x\|)) = \frac{2\sqrt{c}\| - p \oplus_c x\|}{1 - c\| - p \oplus_c x\|^2}. \qquad (63)$$

The other quantity is $\sin(\angle xpw)$ which is minimized when the angle $\angle xpw$ is minimized (because $\angle xpw < \pi/2$ for the hyperbolic right triangle $\Delta xwp$), or, alternatively, when $\cos(\angle xpw)$ is maximized. But, we already have from Eq. (34) that:

$$\cos(\angle xpw) = \frac{\langle -p \oplus_c x, -p \oplus_c w \rangle}{\| - p \oplus_c x\| \cdot \| - p \oplus_c w\|}. \qquad (64)$$

To maximize the above, the constraint on the right angle at $w$ can be dropped because $\cos(\angle xpw)$ depends only on the geodesic $\gamma_{p \to w}$ and not on $w$ itself, and because there is always an orthogonal projection from any point $x$ to any geodesic as stated by Lemma 7. Thus, it remains to find the maximum of Eq. (64) when $w \in \tilde{H}^c_{a,p}$. Using the definition of $\tilde{H}^c_{a,p}$ from Eq. (20), one can easily prove that

$$\{\log^c_p(w) : w \in \tilde{H}^c_{a,p}\} = \{a\}^\perp. \qquad (65)$$

Using that fact that $\log^c_p(w)/\|\log^c_p(w)\| = -p \oplus_c w/\| - p \oplus_c w\|$, we just have to find

$$\max_{z \in \{a\}^\perp} \left( \frac{\langle -p \oplus_c x, z \rangle}{\| - p \oplus_c x\| \cdot \|z\|} \right), \qquad (66)$$

and we are left with a well known Euclidean problem which is equivalent to finding the minimum angle between the vector $-p \oplus_c x$ (viewed as Euclidean) and the hyperplane $\{a\}^\perp$. This angle is given by the Euclidean orthogonal projection whose $sin$ value is the distance from the vector's endpoint to the hyperplane divided by the vector's length:

$$\sin(\angle xpw^*) = \frac{|\langle -p \oplus_c x, \frac{a}{\|a\|} \rangle|}{\| - p \oplus_c x\|}. \qquad (67)$$

It follows that a point $w^* \in \tilde{H}^c_{a,p}$ satisfying Eq. (67) exists (but might not be unique). Combining Eqs. (61),(62),(63) and (67) concludes the proof.

$\square$

## F   Derivation of the Hyperbolic GRU Update-gate

In [24], the authors recover the update/forget-gate mechanism of a GRU/LSTM by requiring that the class of neural networks given by the chosen architecture be invariant to *time-warpings*. The idea is the following.

**Recovering the update-gate from time-warping.**   A naive RNN is given by the equation

$$h(t+1) = \varphi(Wh(t) + Ux(t) + b) \qquad (68)$$

Let's drop the bias $b$ to simplify notations. If $h$ is seen as a differentiable function of time, then a first-order Taylor development gives $h(t + \delta t) \approx h(t) + \delta t \frac{dh}{dt}(t)$ for small $\delta t$. Combining this for $\delta t = 1$ with the naive RNN equation, one gets

$$\frac{dh}{dt}(t) = \varphi(Wh(t) + Ux(t)) - h(t). \qquad (69)$$

As this is written for any $t$, one can replace it by $t \leftarrow \alpha(t)$ where $\alpha$ is a (smooth) increasing function of $t$ called the *time-warping*. Denoting by $\tilde{h}(t) := h(\alpha(t))$ and $\tilde{x}(t) := x(\alpha(t))$, using the chain rule $\frac{d\tilde{h}}{dt}(t) = \frac{d\alpha}{dt}(t)\frac{dh}{dt}(\alpha(t))$, one gets

$$\frac{d\tilde{h}}{dt}(t) = \frac{d\alpha}{dt}(t)\varphi(W\tilde{h}(t) + U\tilde{x}(t)) - \frac{d\alpha}{dt}(t)\tilde{h}(t). \qquad (70)$$

Removing the tildas to simplify notations, discretizing back with $\frac{dh}{dt}(t) \approx h(t+1) - h(t)$ yields

$$h(t+1) = \frac{d\alpha}{dt}(t)\varphi(Wh(t) + Ux(t)) + \left(1 - \frac{d\alpha}{dt}(t)\right)h(t). \tag{71}$$

Requiring that our class of neural networks be invariant to time-warpings means that this class should contain RNNs defined by Eq. (71), *i.e.* that $\frac{d\alpha}{dt}(t)$ can be learned. As this is a positive quantity, we can parametrize it as $z(t) = \sigma(W^z h(t) + U^z x(t))$, recovering the forget-gate equation:

$$h(t+1) = z(t)\varphi(Wh(t) + Ux(t)) + (1 - z(t))h(t). \tag{72}$$

**Adapting this idea to hyperbolic RNNs.** The *gyroderivative* [4] of a map $h : \mathbb{R} \to \mathbb{D}_c^n$ is defined as

$$\frac{dh}{dt}(t) = \lim_{\delta t \to 0} \frac{1}{\delta t} \otimes_c (-h(t) \oplus_c h(t + \delta t)). \tag{73}$$

Using Möbius scalar associativity and the left-cancellation law leads us to

$$h(t + \delta t) \approx h(t) \oplus_c \delta t \otimes_c \frac{dh}{dt}(t), \tag{74}$$

for small $\delta t$. Combining this with the equation of a simple hyperbolic RNN of Eq. (27) with $\delta t = 1$, one gets

$$\frac{dh}{dt}(t) = -h(t) \oplus_c \varphi^{\otimes_c}(W \otimes_c h(t) \oplus_c U \otimes_c x(t)). \tag{75}$$

For the next step, we need the following lemma:

**Lemma 10** (Gyro-chain-rule)**.** *For $\alpha : \mathbb{R} \to \mathbb{R}$ differentiable and $h : \mathbb{R} \to \mathbb{D}_c^n$ with a well-defined gyro-derivative, if $\tilde{h} := h \circ \alpha$, then we have*

$$\frac{d\tilde{h}}{dt}(t) = \frac{d\alpha}{dt}(t) \otimes_c \frac{dh}{dt}(\alpha(t)), \tag{76}$$

*where $\frac{d\alpha}{dt}(t)$ denotes the usual derivative.*

*Proof.*

$$\frac{d\tilde{h}}{dt}(t) = \lim_{\delta t \to 0} \frac{1}{\delta t} \otimes_c [-\tilde{h}(t) \oplus_c \tilde{h}(t + \delta t)] \tag{77}$$

$$= \lim_{\delta t \to 0} \frac{1}{\delta t} \otimes_c [-h(\alpha(t)) \oplus_c h(\alpha(t) + \delta t(\alpha'(t) + \mathcal{O}(\delta t)))] \tag{78}$$

$$= \lim_{\delta t \to 0} \frac{\alpha'(t) + \mathcal{O}(\delta t)}{\delta t(\alpha'(t) + \mathcal{O}(\delta t))} \otimes_c [-h(\alpha(t)) \oplus_c h(\alpha(t) + \delta t(\alpha'(t) + \mathcal{O}(\delta t)))] \tag{79}$$

$$= \lim_{\delta t \to 0} \frac{\alpha'(t)}{\delta t(\alpha'(t) + \mathcal{O}(\delta t))} \otimes_c [-h(\alpha(t)) \oplus_c h(\alpha(t) + \delta t(\alpha'(t) + \mathcal{O}(\delta t)))] \tag{80}$$

$$= \lim_{u \to 0} \frac{\alpha'(t)}{u} \otimes_c [-h(\alpha(t)) \oplus_c h(\alpha(t) + u)] \tag{81}$$

$$= \frac{d\alpha}{dt}(t) \otimes_c \frac{dh}{dt}(\alpha(t)) \qquad \text{(Möbius scalar associativity)} \tag{82}$$

where we set $u = \delta t(\alpha'(t) + \mathcal{O}(\delta t))$, with $u \to 0$ when $\delta t \to 0$, which concludes. $\qquad \square$

Using lemma 10 and Eq. (75), with similar notations as in Eq. (70) we have

$$\frac{d\tilde{h}}{dt}(t) = \frac{d\alpha}{dt}(t) \otimes_c (-\tilde{h}(t) \oplus_c \varphi^{\otimes_c}(W \otimes_c \tilde{h}(t) \oplus_c U \otimes_c \tilde{x}(t))). \tag{83}$$

Finally, discretizing back with Eq. (74), using the left-cancellation law and dropping the tildas yields

$$h(t+1) = h(t) \oplus_c \frac{d\alpha}{dt}(t) \otimes_c (-h(t) \oplus_c \varphi^{\otimes_c}(W \otimes_c h(t) \oplus_c U \otimes_c x(t))). \tag{84}$$

Since $\alpha$ is a time-warping, by definition its derivative is positive and one can choose to parametrize it with an update-gate $z_t$ (a scalar) defined with a sigmoid. Generalizing this scalar scaling by the Möbius version of the pointwise scaling $\odot$ yields the Möbius matrix scaling $\text{diag}(z_t) \otimes_c \cdot$, leading to our proposed Eq. (31) for the hyperbolic GRU.

# G   Experimental details

## G.1   RNN

**Models architecture.**   Our neural network layers can be used in a plug-n-play manner exactly like standard Euclidean layers. They can also be combined with Euclidean layers. However, optimization w.r.t. hyperbolic parameters is different (see below) and based on Riemannian gradients which are just rescaled Euclidean gradients when working in the conformal Poincaré model [21]. Thus, back-propagation can be applied in the standard way.

In our setting, we embed the two sentences using two distinct hyperbolic RNNs or GRUs. The sentence embeddings are then fed together with their squared distance (hyperbolic or Euclidean, depending on their geometry) to a FFNN (Euclidean or hyperbolic, see Sec. 3.2) which is further fed to an MLR (Euclidean or hyperbolic, see Sec. 3.1) that gives probabilities of the two classes (entailment vs neutral). We use cross-entropy loss on top. Note that hyperbolic and Euclidean layers can be mixed, e.g. the full network can be hyperbolic and only the last layer be Euclidean, in which case one has to use $\log_{\mathbf{0}}$ and $\exp_{\mathbf{0}}$ functions to move between the two manifolds in a correct manner as explained for Eq. 24.

**Optimization.**   Our models have both Euclidean (e.g. weight matrices in both Euclidean and hyperbolic FFNNs, RNNs or GRUs) and hyperbolic parameters (e.g. word embeddings or biases for the hyperbolic layers). We optimize the Euclidean parameters with Adam [16] (learning rate 0.001). Hyperbolic parameters cannot be updated with an equivalent method that keeps track of gradient history due to the absence of a Riemannian Adam. Thus, they are optimized using full Riemannian stochastic gradient descent (RSGD) [5, 11]. We also experiment with projected RSGD [21], but optimization was sometimes less stable. We use a different constant learning rate for word embeddings (0.1) and other hyperbolic weights (0.01) because words are updated less frequently.

**Numerical errors.**   Gradients of the basic operations defined above (e.g. $\oplus_c$, exponential map) are not defined when the hyperbolic argument vectors are on the ball border, i.e. $\sqrt{c}\|x\| = 1$. Thus, we always project results of these operations in the ball of radius $1 - \epsilon$, where $\epsilon = 10^{-5}$. Numerical errors also appear when hyperbolic vectors get closer to $\mathbf{0}$, thus we perturb them with an $\epsilon' = 10^{-15}$ before they are used in any of the above operations. Finally, arguments of the $\tanh$ function are clipped between $\pm 15$ to avoid numerical errors, while arguments of $\tanh^{-1}$ are clipped to at most $1 - 10^{-5}$.

**Hyperparameters.**   For all methods, baselines and datasets, we use $c = 1$, word and hidden state embedding dimension of 5 (we focus on the low dimensional setting that was shown to already be effective [21]), batch size of 64. We ran all methods for a fixed number of 30 epochs. For all models, we experiment with both *identity* (no non-linearity) or $\tanh$ non-linearity in the RNN/GRU cell, as well as *identity* or ReLU after the FFNN layer and before MLR. As expected, for the fully Euclidean models, $\tanh$ and ReLU respectively surpassed the *identity* variant by a large margin. We only report the best Euclidean results. Interestingly, for the hyperbolic models, using only identity for both non-linearities works slightly better and this is likely due to the fact that our hyperbolic layers already contain non-linearities by their nature.

For the results shown in Tab. 1, we run each model (baseline or ours) exactly 3 times and report the test result corresponding to the best validation result from these 3 runs. We do this because the highly non-convex spectrum of hyperbolic neural networks sometimes results in convergence to poor local minima, suggesting that initialization is very important.

## G.2   MLR

We use different embedding dimensions : 2, 3, 5 and 10. For the hyperbolic MLR, we use full Riemannian SGD with a learning rate of 0.001. For the two Euclidean models we use ADAM optimizer and the same learning rate. During training, we always sample the same number of negative and positive nodes in each minibatch of size 16; thus positive nodes are frequently resampled. All methods are trained for 30 epochs and the final F1 score is reported (no hyperparameters to validate are used, thus we do not require a validation set). This procedure is repeated for four subtrees of different sizes.

# H More Experimental Investigations

The following empirical facts were observed for both hyperbolic RNNs and GRUs.

We observed that, in the hyperbolic setting, accuracy is often much higher when sentence embeddings can go close to the border (hyperbolic "infinity"), hence exploiting the hyperbolic nature of the space. Moreover, the faster the two sentence norms go to 1, the more it's likely that a good local minima was reached. See figures 3 and 5.

We often observe that test accuracy starts increasing exactly when sentence embedding norms do. However, in the hyperbolic setting, the sentence embeddings norms remain close to 0 for a few epochs, which does not happen in the Euclidean case. See figures 3, 5 and 4. This mysterious fact was also exhibited in a similar way by [21] which suggests that the model first has to adjust the angular layout in the almost Euclidean vicinity of 0 before increasing norms and fully exploiting hyperbolic geometry.

(a) Test accuracy

(b) Norm of the first sentence. Averaged over all sentences in the test set.

Figure 3: PREFIX-30% accuracy and first (premise) sentence norm plots for different runs of the same architecture: hyperbolic GRU followed by hyperbolic FFNN and hyperbolic/Euclidean (half-half) MLR. The X axis shows millions of training examples processed.

(a) Test accuracy

(b) Norm of the first sentence. Averaged over all sentences in the test set.

Figure 4: PREFIX-30% accuracy and first (premise) sentence norm plots for different runs of the same architecture: Euclidean GRU followed by Euclidean FFNN and Euclidean MLR. The X axis shows millions of training examples processed.

(a) Test accuracy

(b) Norm of the first sentence. Averaged over all sentences in the test set.

Figure 5: PREFIX-30% accuracy and first (premise) sentence norm plots for different runs of the same architecture: hyperbolic RNN followed by hyperbolic FFNN and hyperbolic MLR. The X axis shows millions of training examples processed.

## Footnotes

[10]*i.e.* that $\frac{DX}{\partial t} = 0$ for $t \in [0,1]$, where $\frac{D}{\partial t}$ denotes the covariant derivative.