[Reviews · NeurIPS 2018]

Reviewer 1



Thanks to the authors for the detailed response. The new results presented in the rebuttal are indeed convincing, hence I am updating my score to an 8 now. This is with the understanding that these would be incorporated in the revised version of the paper. ========================================= Several works in the last year have explored using hyperbolic representations for data which exhibits hierarchical latent structure. Some promising results on the efficiency of these representations at capturing hierarchical relationships have been shown, most notably by Nickel & Kiela (Nips, 2017). However one big hindrance for utilizing them so far is the lack of deep neural network models which can consume these representations as input for some other downstream task. This paper attempts to solve exactly this important problem. Specifically the paper starts from the Poincare ball model of hyperbolic spaces and developes several standard neural network operations / layers for transformations data lying on this manifold. Section 2 presents the basic definitions of addition and multiplication from the theory of gyrovector spaces and mobius transformations, and derives versions of these for the Poincare ball. Then in section 3 these basic operations are extended to derive the following neural network layers -- (1) multi-class logistic regression, (2) feed forward layers, (3) recurrent neural networks (as well as Gated Recurrent Units). While I am not an expert in Riemannian geometry, the formulations all seem sound. All operations are derived in terms of the curvature, which can be smoothly varied from the Euclidean to Hyperbolic case, however the possible implications of doing so are not discussed. Strengths: 1. The paper tackles an important problem and provides theoretically sound solutions. The results presented here would be highly useful to others working on hyperbolic embedding spaces. 2. The approach is technically sound, and builds on previous theoretical work on gyrovector spaces to provide mathematically sound formulations of hyperbolic neural network layers. This is in contrast to ad-hoc methods which may work empirically but without any clear justification. 3. While the paper is math-heavy, it is well written and the main arguments and claims are easy to follow. Weaknesses: 1. The biggest weakness is that there is little empirical validation provided for the constructed methods. A single table presents some mixed results where in some cases hyperbolic networks perform better and in others their euclidean counterparts or a mixture of the two work best. It seems that more work is needed to clearly understand how powerful the proposed hyperbolic neural networks are. 2. The experimental setup, tasks, and other details are also moved to the appendix which makes it hard to interpret this anyway. I would suggest moving some of these details back in and moving some background from Section 2 to the appendix instead. 3. The tasks studied in the experiments section (textual entailment, and a constructed prefix detection task) also fail to provide any insight on when / how the hyperbolic layers might be useful. Perhaps more thought could have been given to constructing a synthetic task which can clearly show the benefits of using such layers. In summary, the theoretical contributions of the paper are significant and would foster more exciting research in this nascent field. However, though it is not the central focus of the paper, the experiments carried out are unconvincing.

Reviewer 2



This paper takes a principled approach to adapting many of the core operations of neural networks to hyperbolic space. This generalization is done by analogy to the structure of gyro-addition, which gives a convenient pattern for adapting an arbitrary euclidean operation to hyperbolic points while preserving several desirable properties. In particular, this analogy is applied to adapt matrix multiplication, bias addition, and scalar multiplication to hyperbolic space, which, along with a more involved treatment of multinomial logistic regression, give the tools needed to build neural networks. The authors apply this adaptation to hyperbolic GRUs applied to simple tasks. The weakest part of this paper by far is the experiments. The only tasks where the proposed methods outperform the baselines are synthetic. There are also no experiments with hyperbolic MLPs, which I found quite surprising. Most of the paper is devoted to the development of the basic machinery required to build an MLP and I expected to see some experiments in this simple setting before launching into an investigation of recurrent networks (which seem to not work particularly well?) In spite of the disappointing empirical work, I think this paper should still be accepted. Hyperbolic representations are getting a lot of attention recently, and the framework presented in this paper is by far the clearest and most complete application of hyperbolic geometry in neural networks to date. I expect the framework in this paper to be foundational to future applications of hyperbolic geometry to neural networks, to the extent that it has a role to play beyond embedding models. Detailed questions: Why work in the Poincare model instead of one of the other available models? It requires quite some work to derive the exponential and logarithmic maps in this setting, why not work in a model (like the hyperboloid) where they are more easily computed? It would be nice to have some comment on why Poincare (or if it is an arbitrary choice then some indication of that). In Equation 22, what exactly does <-p +_c x, a> mean? As I understand, a \in T_pD and -p +_c x \in D, so what is the <> operator? Have you considered non-linearities that are not derived as the mobius version of standard non-linearities? Perhaps this is worth exploring in light of the comment after Eq 28 in order to make the resulting networks "more hyperbolic"? Following that can you comment on when transforming f(x) to exp(f(log(x)) would not be appropriate? For example, if f(x1, x2,...) is a weighted midpoint operation will exp(f(log(x1), log(x2), …)) compute a hyperbolic weighted midpoint? Is there some characterization or guidance on what properties will and will not be preserved by this analogy? --- After author response: I am happy with the author response, and I still believe this paper should be accepted.

Reviewer 3



A number of recent works in deep learning have been using Hyperbolic Embeddings to represent features in Neural Networks. One main advantage of these representations is that they use a more natural geometry to represent hierarchical data such as trees. This work seeks a generic theoretical grounding as well as practical technique for the use of Hyperbolic Geometry at various layers of a neural network. Pros: * Overall well written and clear; * A lot of the ideas are well motivated, and the authors took care to give intuitive justifications for many of their choices; * Hyperbolic geometry has been used in multiple machine learning papers recently, and this new paper is a good attempt at proposing a general framework; * I found the use of parallel transport to define biases in hyperbolic networks, particularly nice and clever. Cons: * experiments are very limited in details (in the main part of the paper), since this is mostly a theory paper, I see this as a minor point; * the introduction points to a number of recent works using hyperbolic geometry. It would be good to have a bit more details about differences between these approaches, and the one proposed in the paper. Minor/easily fixed comments: * A number of things about the definition of a geodesic on lines 70-71 are not quite right. Firstly, there are many paths of minimal lengths between points on a Riemannian manifold that are not geodesics (in the Riemannian sense). Indeed, the geodesic paths are only the ones that are parameterised proportionally to arc length. For ex, on the real line R, the path $\gamma(t) = t^2$ is of minimal length between $\gamma(0)$ and $\gamma(1)$, but it is not a geodesic. Secondly, on a general Riemannian manifold, there can be paths between two points that are not of minimal length, but are geodesics nonetheless. For example, on a sphere, for two points close to each other, there are two geodesics along the great circle through these two points. What is true, is that geodesics are locally of minimal length. On the Poincaré ball, there is always only one geodesic that joins any two points. The author should make it clear that they use this very particular property, and that it is not clear in general. * On line 72, parallel transport is introduced as a transformation that depends only on the starting and end point $P_{x\rightarrowy}$, but parallel transport is a transformation that does depend on the whole path. In this paper, the authors implicitly always choose the unique geodesic between two points on the Poincaré ball. This should be made clear. * Definition of exponential map on line 76 is unclear: the paragraph makes it look like it is defined only for tangent vectors of norm 1. Which is incorrect. * On line 106, the sum of x and 0 should be x, not 0. * Around line 113, I would suggest adding that $(-1)\bigotimes_c x = -x$. * In Theorem 4, here as well the authors should make it clear that this is for parallel transport along a geodesic. * In the 'Reformulating Euclidean MLR' section, from line 177, the first proposed parameterisation of hyperplanes as $H_{a,b}$ is not actually used. Only the second one (which is a very natural parameterisation), $\tilde{H}_{a, p}$, is used. Introducing both parameterisations is confusing. Removing the first one would also save space. More important comments: * All hyperbolic balls of the same dimension are isomorphic. Indeed the map $x \mapsto \frac{\sqrt{c}}{\sqrt{c'}} x$ is an isometric diffeomorphism that transforms $\bigoplus_c$ into $\bigoplus_{c'}$, and so on. This is an important point. And using this point would help declutter many of the formulas: the $c$ subscript could be dropped in most places, since the equivalent formula for a general $c$ can easily be deduced from the case $c=1$. The only time that $c$ is useful in the paper is when the authors consider what happens when $c$ goes to $0$. Note also that in the experiments, only $c=1$ is actually used. * Regarding the section on hyperparameters, at line 552, could the authors clarify what they mean by "the hyperbolic geometry is more pronounced at the ball border"? The geometry is constant across the whole Poincaré ball. In particular, all points are equal and for any point, there exists an isometry of the ball that brings this point to 0. The only particular thing for the border is due to the particular parameterisation of the Poincaré ball: as we approach the boundary (in the sense of Euclidean metric, since in terms of hyperbolic metric, the border is always infinitely far), we lose precision in our compute. ===================== Thanks to the author for their replies. I have read them, and indeed think this is very good submission. Please, do point out the isomorphisms between the balls with various 'c'. This is the main confusing point in the paper.